# The Changes of Amino-Acid Metabolism between Wheat and Rice during Early Growth under Flooding Stress

**DOI:** 10.3390/ijms25105229

**Published:** 2024-05-11

**Authors:** Setsuko Komatsu, Mayu Egishi, Toshihisa Ohno

**Affiliations:** Faculty of Life and Environmental Sciences, Fukui University of Technology, Fukui 910-8505, Japan

**Keywords:** wheat, rice, flooding stress, gamma-aminobutyric acid shunt, anaerobic/aerobic metabolism

## Abstract

Floods induce hypoxic stress and reduce wheat growth. On the other hand, rice is a semi-aquatic plant and usually grows even when partially submerged. To clarify the dynamic differences in the cellular mechanism between rice and wheat under flooding stress, morphological and biochemical analyses were performed. Although the growth of wheat in the early stage was significantly suppressed due to flooding stress, rice was hardly affected. Amino-acid analysis revealed significant changes in amino acids involved in the gamma-aminobutyric acid (GABA) shunt and anaerobic/aerobic metabolism. Flood stress significantly increased the contents of GABA and glutamate in wheat compared with rice, though the abundances of glutamate decarboxylase and succinyl semialdehyde dehydrogenase did not change. The abundance of alcohol dehydrogenase and pyruvate carboxylase increased in wheat and rice, respectively. The contents of aspartic acid and pyruvic acid increased in rice root but remained unchanged in wheat; however, the abundance of aspartate aminotransferase increased in wheat root. These results suggest that flooding stress significantly inhibits wheat growth through upregulating amino-acid metabolism and increasing the alcohol-fermentation system compared to rice. When plant growth is inhibited by flooding stress and the aerobic-metabolic system is activated, GABA content increases.

## 1. Introduction

Climate change, which is one of the challenges facing modern agriculture, is exacerbated by global population growth and soil-quality deterioration [1]. Environmental stress has increased in frequency in recent years due to climate change, which occurs via stressors in combination rather than individually causing large-scale crop losses around the world [2]. In nature, multiple stressors occur simultaneously, such as a combination of drought and heat stress, or sequentially, such as a drought followed by a flood [3]. On the other hand, adaptive responses to artificially applied single-factor stresses under laboratory conditions have been described for multiple types of crops [4]. Particularly, in the context of flooding, understanding of several stress responses at the molecular level is still ambiguous, which is a disadvantage considering that flooding often occurs as a co-stress with salinity [5]. Because flooding stress is based on initial fluctuations and also induces secondary changes, its mechanism is not completely understood.

Rice is a semi-aquatic plant, which usually grows even when partially submerged; however, excessively prolonged waterlogging can limit plant growth and grain yield [6]. Rice grows well in standing water, but most varieties will die if completely submerged for more than 3 days [7]. Rice uses two strategies to survive submergence, which are quiescence and escape. Ethylene accumulation in plant cells as a result of submersion induces *Submergence 1A* (*SUB1A*) in genotypes which possess this gene [8]. *SUB1A* induced a quiescence status in submerged rice plants, reducing growth and the use of carbohydrates. *SUB1A*, which is an ethylene-response factor (ERF), confers submergence tolerance to rice by restricting shoot elongation during the inundation period. It is proposed to limit shoot growth by regulating gibberellic acid signaling [9]. Rice can preserve carbon reserves, allowing rice plants to re-grow even when the water recedes.

On the contrary, an escape strategy is exploited in another group of rice varieties, such as deepwater rice, which grows very rapidly in response to submergence [10]. Ethylene trapped by water surrounding the rice plant induces *SNORKEL* genes, which cause the stem to elongate rapidly. This enables the plant to keep its leaves above the water surface, allowing oxygen to be transported from the air to the submerged parts of the plant. *SNORKEL1* and *SNORKEL2* belong to the ERFVII transcription-factor subgroup, which triggers a gibberellin-dependent internode elongation [10]. Among cereals, rice has the remarkable ability to germinate under water [11] and elongates coleoptiles with a snorkel-like function [12]. The ability to break down starch in the absence of oxygen is important to provide energy for organs and their growth.

In the case of wheat, the seedling stage is one of the critical stages where post-emergence waterlogging has the most negative impact on wheat yield [13]. Wheat yield losses due to flooding can range from 10% to more than 50% [14]; however, it depends on waterlogging duration, wheat genotype, growth stage, soil type, and agricultural management. Water near the soil surface floods all the tissues of wheat seeds/seedlings, including the coleoptiles [15,16]. With exogenous sugar supply, it has been shown that 84% of seeds germinate, whereas only slight root elongation and no coleoptile growth has been observed in the absence of hypoxia [17]. Moreover, short waterlogging periods for 3 days can result in long-term detrimental effects on both the plant growth and grain yield of wheat [18]. In contrast to rice, wheat seeds cannot germinate under anoxic conditions. Specifically, wheat seeds under hypoxia are unable to decompose starch into sugar, nor germinate.

Epigenetic changes are associated with abiotic stress in crops. Heat stress increases DNA methylation levels in heat-sensitive rice but decreases it in heat-tolerant rice [19]. The total DNA methylation in sesame increases under drought stress but decreases under flooding stress [20]. In wheat, DNA demethylation significantly increases waterlogging-related gene expression in tolerant genotypes under hypoxic stress [21]. Both waterlogging and half submergence increase the total DNA methylation level, but this is decreased under full submergence in wheat [22]. However, the potential mechanisms of epigenetic regulation under flooding stress remain largely unknown in rice and wheat.

Plants can cope with flooding conditions by embracing an orchestrated set of morphological adaptations and physiological adjustments, which are regulated by an elaborate hormonal signaling network [23]. In wheat seedlings, the upregulation of *TDC*, *YUC1*, and *PIN9* involved in auxin biosynthesis and transport contribute to high levels of auxin, which is required for nodal root induction during hypoxia [24]. Regarding salicylic acid enhancement of wheat tolerance to waterlogging, it has been shown that salicylic acid promotes formation of axile roots independent from ethylene, but its effect on adventitious rooting is dependent on ethylene [25]. The hormonal signaling networks are also important in wheat under flooding stress, but they are less clearly understood than in rice.

Metabolites sensitive to oxygen limitation include nitrogen compounds, which are closely linked to the utilization of pyruvate and its further association with the metabolism of amino acids [26]. An increased content of amino acids has been observed in rice coleoptiles during anoxia [27]. The tolerant cultivar of wheat was characterized by higher levels of amino acids [28]. The accumulation of amino acids associated with the gamma-aminobutyric acid (GABA) shunt and derived from the metabolites of glycolysis has been detected during oxygen deprivation in the majority of plant species regardless of their resistance to waterlogging/submergence. Understanding the mechanisms by which wheat copes with unexpected flooding is important for developing new flooding-tolerant wheat cultivars. In this study, to clarify the dynamic differences in cellular mechanisms between rice and wheat under flooding stress, morphological parameters were measured. Based on the results, amino-acid analysis was performed. Following this result, biochemical and enzymatic analyses were conducted.

## 2. Results

### 2.1. Morphological Changes of Rice and Wheat Treated with Flooding Stress

To investigate the effect of flooding stress on rice and wheat, morphological analysis was performed. Before sowing, rice seeds were allowed to absorb water in a Petri dish for 7 days. The seeds of rice and wheat were sown, and 3-day-old plants were flooded for 2 days (Figure 1). The first leaf of rice and wheat were already formed and their seedling sizes were about the same before the flood. However, the rice was slightly smaller, although rice seeds were allowed to absorb water in a Petri dish for 7 days before sowing (Figure 2A). As morphological parameters, leaf length (Figure 2B), leaf-fresh weight (Figure 2C), main-root length (Figure 2D), and total-root fresh weight (Figure 2E) were measured. All parameters decreased in wheat under flooding. However, only total-root fresh weight decreased and other parameters did not change in rice under the same condition (Figure 2). Based on the morphological results, the roots of rice and wheat were used for amino-acid analysis.

### 2.2. The Changes of Amino Acids in the Roots of Rice and Wheat Treated with Flooding Stress

To understand the differences of amino-acid metabolism in the roots of wheat and rice during early growth under flooding stress, the contents of amino acids were analyzed using an automatic amino-acid analyzer. NH_3_ and 25 components related to amino acids were identified in wheat and rice (Appendix A; Figure 3). Based on amino-acid analysis (Appendix A), NH_3_ and 21 amino acids, except PEA, a-ABA p-Ser, and MEA, were mapped on amino-acid metabolism using the Kyoto Encyclopedia of Gene and Genomes (KEGG) database (Figure 4). NH_3_ slightly increased in rice but decreased in wheat under flooding stress. Among the 21 amino acids, the contents of 19 amino acids increased in rice and wheat under flooding stress compered to control; especially, the contents of GABA, alanine (Ala), and lysine (Lys) significantly increased in both plants under the same condition. Between rice and wheat, the contents of GABA, Lys, Ala, and isoleucine (Ile) in wheat were higher than those in rice under flooding stress. Oppositely, the contents of tyrosine (Tyr), glycine (Gly), serine (Ser), and glutamine (Gln) in wheat were lower than those in rice under flooding stress. The contents of aspartic acid (Asp) and methionine (Met) decreased in wheat under flooding stress compared to control; however, they slightly increased in rice under the same condition (Figure 3 and Figure 4). These results indicated that amino-acid metabolism was significantly affected in wheat and rice by flooding stress. Based on the results of amino-acid analysis, confirmation analyses were performed.

### 2.3. Immunoblot Analysis of Proteins Related to GABA Shunt and Fermentation in Rice and Wheat under Flooding Stress

To confirm the results of amino-acid analysis in rice and wheat under flooding stress, the abundance of proteins related to GABA shunt as well as fermentation were analyzed using immunoblot analysis. Three-day-old plants were flooded with water for 2 days. Proteins extracted from roots and leaves were separated on SDS-polyacrylamide gel by electrophoresis and transferred onto membrane. The membrane was cross-reacted with the primary antibodies. A staining pattern with Coomassie-brilliant blue was used as a loading control (Figure 5A and Appendix A). The integrated densities of bands were calculated using ImageJ software with triplicated immunoblot results (Figure 5, Figure 6 and Appendix A).

As proteins related to GABA shunt, the abundances of glutamate decarboxylase (GAD) and succinyl semialdehyde dehydrogenase (SSADH) were selectively analyzed using immunoblot analysis (Figure 5). The abundances of GAD and SSADH did not significantly change in rice and wheat under flooding stress (Figure 5B,C, Appendix A). As proteins related to aerobic and anaerobic metabolism, the abundances of alcohol dehydrogenase (ADH) and pyruvate carboxylase were selectively analyzed using immunoblot analysis (Figure 6). The abundance of ADH increased in wheat root under flooding stress; however, that of pyruvate carboxylase increased in both root and leaf of rice under flooding stress (Figure 6A,B, Appendix A). Because the increase and decrease in Asp level in rice and wheat were opposite, the abundance of aspartate aminotransferase was analyzed using immunoblot analysis (Figure 6C). The abundance of aspartate aminotransferase increased in both root and leaf of wheat under flooding stress; however, it did not change in rice even if it was under flooding stress (Figure 6C and Appendix A).

### 2.4. The Contents of GABA, Glutamic Acid, Pyruvic Acid, and Asp in Rice and Wheat under Flooding Stress

To confirm the effect on GABA shunt and the anaerobic-metabolic system of flooding stress in rice and wheat, the contents of GABA, glutamic acid (Glu), pyruvic acid, and Asp were analyzed. Three-day-old rice and wheat were flooded with water for 2 days. Root and leaf were homogenized with phosphate-buffered saline. After removal of proteins, the contents of GABA, Glu, pyruvic acid, and Asp were analyzed (Figure 7).

The content of GABA significantly increased in the root and leaf of rice and wheat under flooding stress; these changes were more pronounced in wheat than in rice (Figure 7A). The content of Glu significantly increased in root treated with flooding stress; furthermore, this increase was significant in wheat leaf under flooding stress (Figure 7B). The content of Asp and pyruvic acid significantly increased in rice root treated with flooding stress; however, it did not change in wheat even if it was under flooding stress (Figure 7C,D).

## 3. Discussion

### 3.1. Flooding Stress Significantly Suppresses the Growth of Wheat Compared to Rice at the Early-Growth Stage

Rice seedlings are suited to stagnant conditions because they have well-established aerenchyma, which is enabling oxygen transportation through the roots [29]. On the other hand, the seedling stage of wheat is one of the critical stages where post-emergence waterlogging has the most negative impact on grain yield [13]. In this study, wheat growth was significantly reduced under flooding; however, there was no effect on rice growth under the same condition (Figure 2). Waterlogging impairs oxygen circulation in the rhizosphere, limiting the oxygen availability to the roots. When plants are completely submerged, oxygen is completely depleted. This is one of the stress factors which cause serious harm to plants [26,30]. Oxygen deficiency not only affects plant growth, development, and distribution in terrestrial/aquatic ecosystems, but also affects grain yield losses around the world [31]. The lack of molecular oxygen leads to the inhibition of aerobic respiration, which causes energy starvation and the acceleration of glycolysis to fermentations.

Root growth is a well-known outcome of polar-auxin transport and canalization. Amino acids and auxin need to be transported across cell membranes to exert their functions in various organs, a process which relies on specific carrier proteins on every membrane, mainly amino-acid transporter proteins [32]. Two families of amino-acid transporter proteins were identified, which are amino acid/auxin permease and the amino-acid polyamine choline gene family [33]. Amino acid/auxin permease protein is an enzyme which mediates the movement of a variety of amino acids and auxin into and out of cells [34], and participates in regulating the transmembrane structure of amino acids, the long-distance transport of amino acids in the body, and other life processes [35]. Under saline-alkali stress, amino acid/auxin permease was differentially expressed between a salt-alkali-tolerant millet variety and a salt-alkali-sensitive millet variety [36]. The reason why amino-acid analysis was adopted in this study is that changes in the number of amino acids in plants under flooding stress are related to various metabolisms. For example, research has shown that amino acids related to glycolysis, including the phosphoglycerate family (Ser and Gly), shikimate family (Phe, Tyr, and Trp), and pyruvate family (Ala, Leu, and Val) were significantly elevated. Members of the Asp family (Asn, Lys, Met, Thr, and Ile) and the Glu family (Glu, Pro, Arg, and GABA) were accumulated as well [19]. This study focuses on systemic metabolic changes revealed by amino-acid metabolism during early growth in rice and wheat.

### 3.2. Alcohol Fermentation Enhances in Wheat Compared to Rice at the Early-Growth Stage under Flooding Stress

Enhancement of glycolysis for ATP synthesis and NAD^+^ regeneration by alcohol fermentation are the most important metabolic mechanisms developed during the aerobic to anaerobic metabolic transition, and changes in carbohydrate profile are vital for metabolic adaptation to oxygen deprivation [4]. In this study, the abundance of ADH increased in wheat root compared to rice root under flooding stress; however, that of pyruvate carboxylase increased in rice compared to wheat under the same condition (Figure 6). ADH can interconvert ethanol and acetaldehyde or other corresponding alcohol/aldehyde pairs in the alcohol-fermentation pathway, which has low toxicity to organisms, and regenerate nicotinamide-adenine dinucleotide to maintain glycolysis metabolism [37]. The relationship between ADH activity and resistance to oxygen deficiency has been well demonstrated in previous research. On the other hand, phospho*enol*pyruvate carboxylase, which is essential for plant individuals, catalyzes the irreversible beta-carboxylation of phospho*enol*pyruvate by incorporating HCO_3_^−^ to yield oxaloacetate [38]. Plant-phospho*enol*pyruvate carboxylase plays a role in nitrogen metabolism, fatty acid biosynthesis, and respiration [39]. Additionally, it refixes CO_2_ released through respiration and acts in abiotic stress adaptation [40]. While rice can replenish oxygen through aerobic tissues due to flooding stress during the early stages of growth, wheat may need to switch to an anaerobic-metabolic system during the same point.

### 3.3. GABA Is More Accumulated in Wheat Compared to Rice under Flooding Stress

Metabolites sensitive to oxygen limitation include nitrogen compounds, which are associated with pyruvate utilization and amino-acid metabolism [26]. In this study, the accumulation of amino acids related with Glu metabolism, GABA shunt, and amino-acids desorption from glycolysis metabolites, was identified under flooding stress (Figure 4). Excess Glu enters a bypass branch of the Krebs cycle called the GABA shunt and is decarboxylated via the production of GABA [4], which is converted to succinic semialdehyde [41]. This reaction is catalyzed by GABA transaminase and is coupled to the synthesis of Ala from pyruvate. Succinic semialdehyde is converted to succinate and then to fumarate. The oxidation of succinic semialdehyde consumes NAD^+^ and is inhibited by the lack of oxygen [42], whereas the conversion of GABA to succinate is reactivated upon reaeration [43]. In the present study, GABA accumulation in wheat root was higher than in rice root (Figure 3), indicating that wheat is more susceptible to flooding stress than rice.

Oxaloacetate is produced by aspartate aminotransferase [44]. To maintain this pathway, Asp and Gln pools are depleted [26]. Malate is either decarboxylated by malic enzyme to form pyruvate or reacts via fumarate to form succinate in the reversed dicarboxylic portion of the Krebs cycle [44]. In this study, Asp slightly increased in wheat but significantly increased in rice under flooding stress (Figure 6 and Figure 7). The promoter analysis of upregulated genes revealed that the production of oxaloacetate from Asp via Asp oxidase, energy processes, and the lignification process were likely under abscisic-acid control [45]. The present study, along with previous findings, indicates that the reason Asp is so low in wheat is because aspartate aminotransferase acts to produce oxaloacetate under flooding stress.

## 4. Materials and Methods

### 4.1. Plant Material and Treatment

Seeds of wheat (*Triticum aestivum* L. cultivar Nourin 61) and rice (*Oryza sativa* L. cultivar Nipponbare) were sterilized with 2% sodium-hypochlorite solution and rinsed twice with water. Before sowing, rice seeds were allowed to absorb water in a Petri dish for 7 days. The seeds of rice and wheat were sown in 400 mL of silica sand in a seedling case. A total of 20 seeds were sown evenly in each seedling case. Plants were grown in a growth chamber with white fluorescent light (12 h light of 200 µmol m^−2^ s^−1^ and 12 h dark photoperiod) with 60% humidity at 25 °C. To induce flooding stress, water was added to 5 cm above the sand surface, and 3-day-old plants were soaked for 2 days. The sowing of seeds was carried out on different days for making biological replicates. Three independent experiments were performed as biological replicates for all experiments (Figure 1).

### 4.2. Amino-Acid Analysis

A portion (500 mg) of each sample was excised into small pieces and put into a mortar and pestle. Each portion was ground in 500 µL of phosphate-buffered saline, which contained 137 mM NaCl, 2.7 mM KCl, 10 mM Na_2_HPO_4_, and 1.76 mM KH_2_PO_4_ (pH 7.4). The suspension was centrifuged twice at 16,000× *g* for 20 min at 4 °C. The final supernatant was mixed with the same amount of 3% sulfosalicylic acid and centrifuged at 16,000× *g* for 20 min at 4 °C to remove precipitated proteins. After filtration, the amino-acid concentrations were analyzed with ninhydrin reagent using a fully automatic amino-acid analyzer (JLC-500/V; JEOL, Tokyo, Japan). The unit “nmol” shows amino-acid concentration per mg of root-fresh weight.

### 4.3. Immunoblot Analysis

A portion (500 mg) of each sample was excised into small pieces and put into a mortar and pestle. Each portion was ground in 500 µL of lysis buffer, which contained 50 mM Tris-HCl, 150 mM NaCl, 1% Nonidet-P40, 0.1% SDS, and protease inhibitor (Nacalai Tesque, Kyoto, Japan). The suspension was centrifuged twice at 16,000× *g* for 10 min at 4 °C. Proteins were reacted with XL-Bradford (Apro Science, Tokushima, Japan) and absorbance was measured at 595 nm. A standard curve to determine protein concentration was prepared with bovine serum albumin [46].

Extracted proteins were added in a SDS-sample buffer, which contained 60 mM Tris-HCl (pH 6.8), 2% SDS, 10% glycerol, and 50 mM dithiothreitol (Bio-Rad, Hercules, CA, USA) [47]. Proteins (10 µg) were separated by electrophoresis on a 10% SDS-polyacrylamide gel and transferred onto a polyvinylidene difluoride membrane using a semidry transfer blotter. The blotted membrane was blocked for 5 min in Bullet Blocking One reagent (Nacalai Tesque). After washing, as the primary antibodies, anti-GAD (Bioworld Technology, St. Louis Park, MN, USA), SSADH (abcam, Cambridge, UK), ADH [48], pyruvate carboxylase (Proteintech, Rosemont, IL, USA), and aspartate aminotransferase (Aviva Systems Biology, San Diego, CA, USA) antibodies were used for a 30 min incubation. After washing, as the secondary antibody, anti-rabbit IgG conjugated with horseradish peroxidase (Bio-Rad) was used for a 30 min incubation. The signals were detected using 3,3′,5,5′-tetramethylbenzidine solution (Nacalai Tesque). Coomassie-brilliant blue staining was used as a loading control. The integrated densities of bands were calculated using Image J software (version 1.53e with Java 1.8.0_172, 64 bit; National Institutes of Health, Bethesda, MD, USA).

### 4.4. Assay of the Contents of Pyruvic Acid, GABA, Glutamic Acid, and Aspartic Acid

A portion (250 mg) of each sample was excised into small pieces and put into a mortar and pestle. Each portion was ground in 250 µL of phosphate-buffered saline, which contained 137 mM NaCl, 2.7 mM KCl, 10 mM Na_2_HPO_4_, and 1.76 mM KH_2_PO_4_ (pH 7.4). The suspension was centrifuged twice at 16,000× *g* for 20 min at 4 °C. The supernatant was used for the following analyses.

Contents of pyruvic acid were analyzed using the Amplite Colorimetric Pyruvate Assay Kit (AAT Bioquest, Sunnyvale, CA, USA). Pyruvate standards, blank controls, and samples were prepared with 50 µL according to the layout provided by the company. Working solution (50 µL) was added to tubes of pyruvate standard, blank control, and samples to make the total pyruvate assay volume of 100 µL. The reaction mixture was incubated for 30 min to 1 h. The absorbance increase was monitored at 575 nm.

Contents of GABA were analyzed using the GABA enzymatic assay kit (Enzyme Sensor, Tsukuba, Japan), which consists of two solutions: solution I containing 10 U/mL ascorbate oxidase, 0.8 U/mL glutamate oxidase, 1200 U/mL catalase, 10 U/mL peroxidase, and 0.8 mM 4-aminoantipyrine; and solution II containing 2 U/mL GABA-aminotransferase, 0.8 mM N-ethyl-N-(2-hydroxy-3-sulfopropyl)-3-methylaniline sodium salt, 1 mM sodium 2-oxoglutarate, 2 mM pyridoxal phosphate, and 0.09% sodium azide. For measurement, 0.5 mL of solution I was added to 50 μL of sample solution containing GABA and incubated for 10 min at 30 °C. After incubation, 0.5 mL of solution II was added and incubation was continued for 20 min at 30 °C. After incubation, each assay was measured at absorbance at 555 nm. GABA content was determined by referring to the calibration curves [49].

The glutamic-acid enzymatic assay kit (Enzyme Sensor) consisted of two solutions, which were GLU-A and GLU-B. The procedure used was the same as the method above. The aspartic-acid enzymatic assay kit (Enzyme Sensor) also consisted of two solutions, which were ASN-A and ASN-B. The procedure used was the same as the method above.

### 4.5. Statistical Analysis

The statistical significance of the data was analyzed using Student’s *t*-test. A *p*-value of less than 0.05 was considered statistically significant.

## 5. Conclusions

Wheat is a highly adaptable crop and is widely grown around the world. However, its growth is hampered by flooding, which causes hypoxic stress. Rice, on the other hand, is a semi-aquatic plant and usually grows even when partially submerged. Flood stress significantly suppressed the early growth of wheat, but had little effect on rice. To clarify the dynamic differences in cellular mechanisms between rice and wheat under flooding stress, amino-acid analysis was performed. Amino-acid analysis revealed significant changes in amino acids involved in the GABA shunt and anaerobic/aerobic metabolism. These results were confirmed by biochemical and immunoblot analyses. The main findings are as follows (Figure 8): (i) flood stress significantly increased the contents of GABA and glutamate in wheat compared to rice, though the abundances of glutamate decarboxylase and succinyl semialdehyde dehydrogenase did not change; (ii) the abundance of alcohol dehydrogenase and pyruvate carboxylase increased in wheat and rice, respectively; (iii) the abundances of aspartic acid and pyruvic acid increased in rice root but remained unchanged in wheat; (iv) the abundance of aspartate aminotransferase increased in wheat root. These results suggest that flooding stress significantly inhibits wheat growth through upregulating amino-acid metabolism and increasing the alcohol-fermentation system compared to rice. When plant growth is inhibited by flooding stress and the aerobic-metabolic system is activated, GABA content increases.

## Figures and Tables

**Figure 1 ijms-25-05229-f001:**
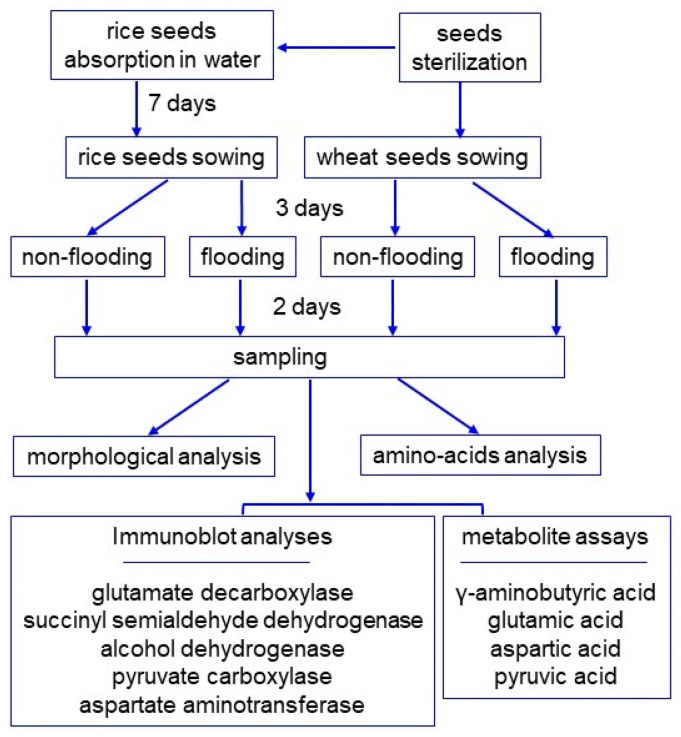
Experimental design for investigation of the effect on rice and wheat of flooding stress. Seeds were sterilized with 2% sodium-hypochlorite solution and rinsed twice with water. Before sowing, rice seeds were allowed to absorb water for 7 days. The seeds of rice and wheat were sown, and 3-day-old plants were flooded for 2 days. Rice and wheat seedlings were analyzed with morphological, amino acids, immunoblot, and metabolite analyses. All experiments were performed with three independent biological replicates.

**Figure 2 ijms-25-05229-f002:**
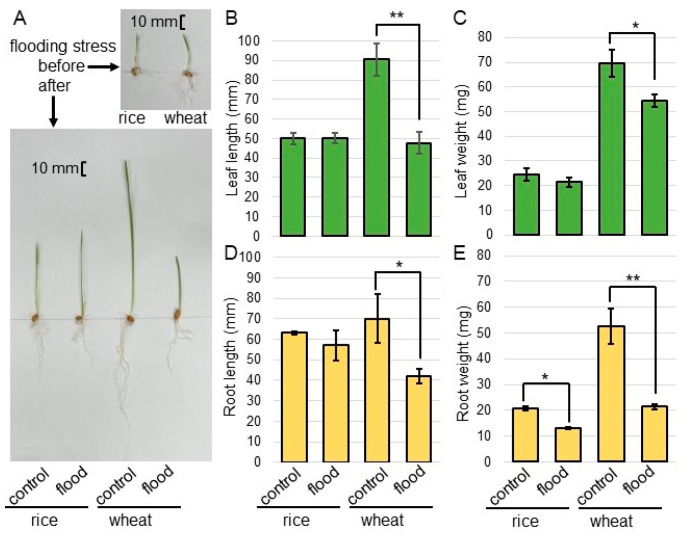
The morphological changes in rice and wheat under flooding stress. Seeds of wheat and rice were sterilized with sodium-hypochlorite solution and rinsed twice with water. Before sowing, rice seeds were allowed to absorb water for 7 days. The seeds of rice and wheat were sown, and 3-day-old plants were flooded for 2 days. Photographs show the plants before and after flooding stress (**A**). As morphological parameters, leaf length (**B**), leaf-fresh weight (**C**), main-root length (**D**), and total-root fresh weight (**E**) were measured. The bar in the left panel indicates 10 mm. The data are presented as mean ± SD from three independent biological replicates. Asterisks indicate significant changes in the flooding compared to control according to Student’s *t*-test (**, *p* < 0.01; *, *p* < 0.05).

**Figure 3 ijms-25-05229-f003:**
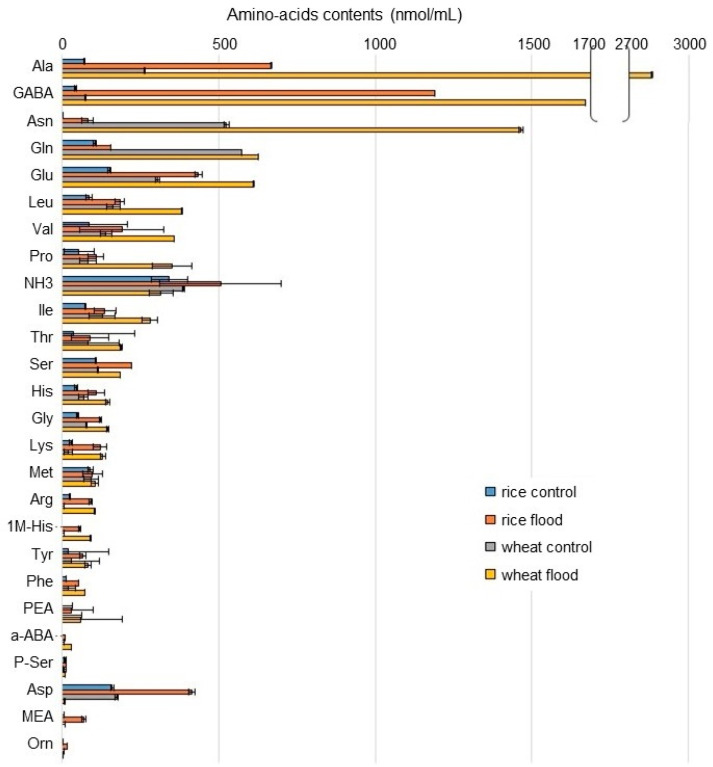
The contents of amino acids in the roots of rice and wheat under flooding stress. The seeds of rice and wheat were sown, and 3-day-old plants were treated with or without flood for 2 days. Roots were collected from seedlings. The abundance of amino acids was analyzed using an automatic amino-acid analyzer. The data are presented as mean ± SD from three independent biological replicates (Appendix A).

**Figure 4 ijms-25-05229-f004:**
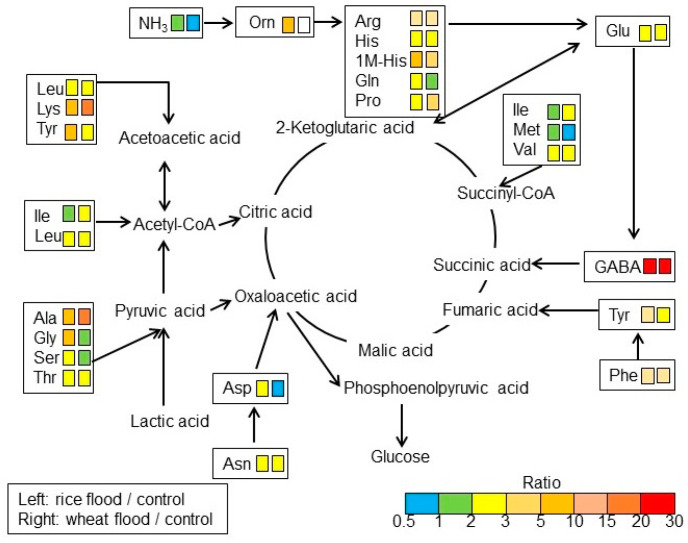
A mapping of altered amino acids to amino-acid metabolism in the roots of rice and wheat under flooding stress. Identified amino acids were mapped onto a pathway according to the KEGG database. Based on amino-acid analysis (Appendix A), 21 amino acids and NH_3_, except PEA, a-ABA p-Ser, and MEA, were mapped. Differences in color indicate the different ratio ranges of the quantities of metabolites, which were calculated by dividing the content in flooded rice or wheat by the content in corresponding non-flooded plants. Uncolored box indicates amino acids that could not be identified. Each set of two boxes shows that the left is “flood/control in rice” and the right is “flood/control in wheat”.

**Figure 5 ijms-25-05229-f005:**
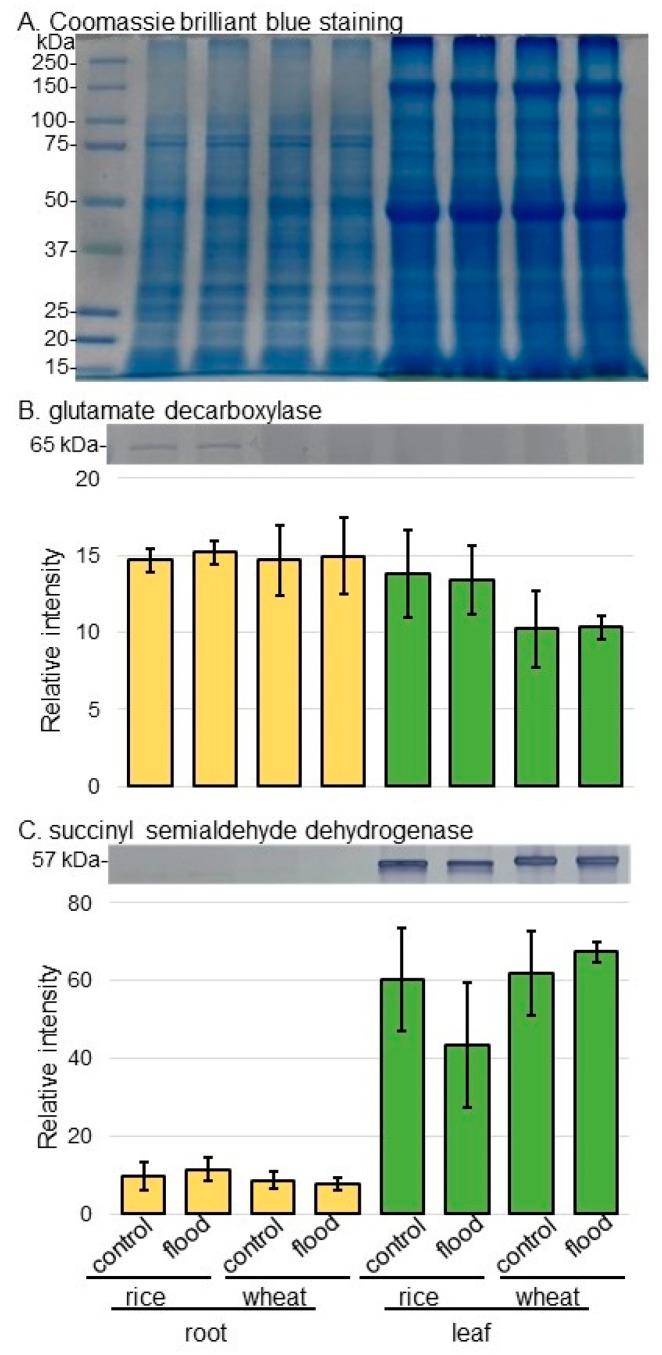
Immunoblot analysis of GAD and SSADH in rice and wheat under flooding stress. Proteins extracted from the root and leaf of rice and wheat seedlings were separated on SDS-polyacrylamide gel by electrophoresis and transferred onto membranes. The membranes were cross-reacted with anti-GAD antibody or anti-SSADH antibody. A staining pattern with Coomassie-brilliant blue was used as a loading control (**A**). The integrated densities of bands of GAD (**B**) and SSADH (**C**) were calculated using ImageJ software. The data are presented as mean ± SD from three independent biological replicates (Appendix A).

**Figure 6 ijms-25-05229-f006:**
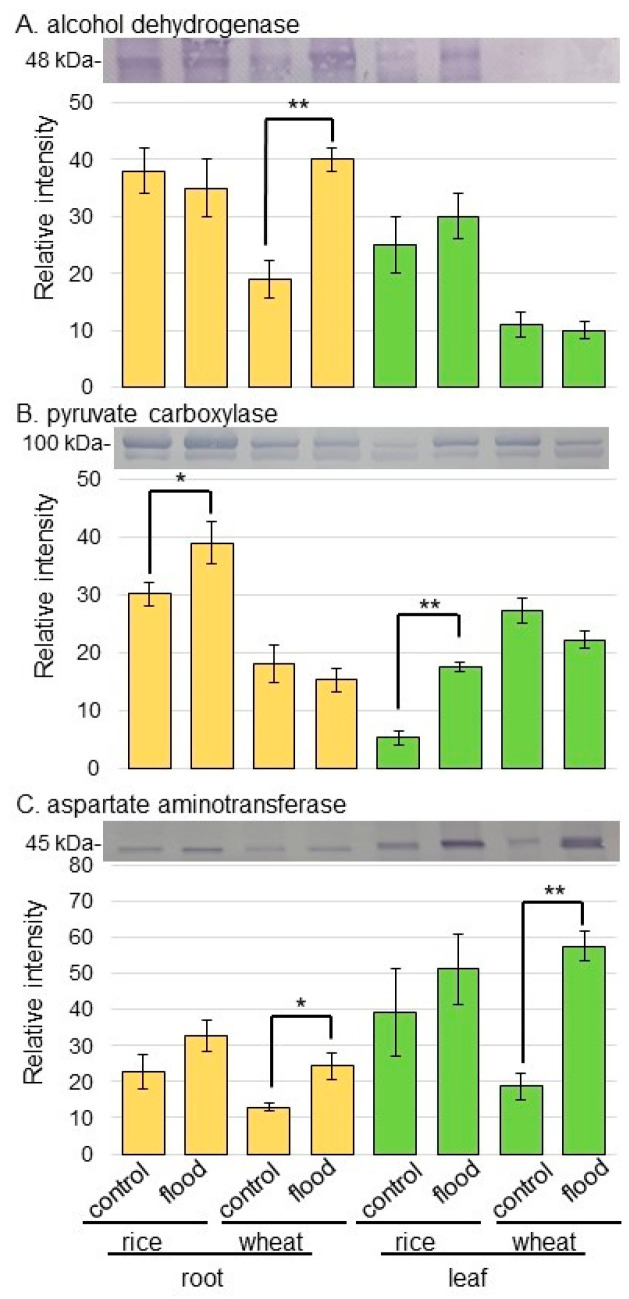
Immunoblot analysis of ADH, pyruvate carboxylase, and aspartate aminotransferase in rice and wheat under flooding stress. Proteins extracted from the root and leaf of rice and wheat seedlings were separated on SDS-polyacrylamide gel by electrophoresis and transferred onto membranes. The membranes were cross-reacted with anti-ADH antibody, anti-pyruvate carboxylase antibody, and anti-aspartate aminotransferase antibody. A staining pattern with Coomassie-brilliant blue was used as a loading control. The integrated densities of bands of ADH (**A**), pyruvate carboxylase (**B**), and aspartate aminotransferase (**C**) were calculated using ImageJ software. The data are presented as mean ± SD from three independent biological replicates (Appendix A). Asterisks indicate significant changes in the relative intensity of signal band in the flooding compared to control according to Student’s *t*-test (**, *p* < 0.01; *, *p* < 0.05).

**Figure 7 ijms-25-05229-f007:**
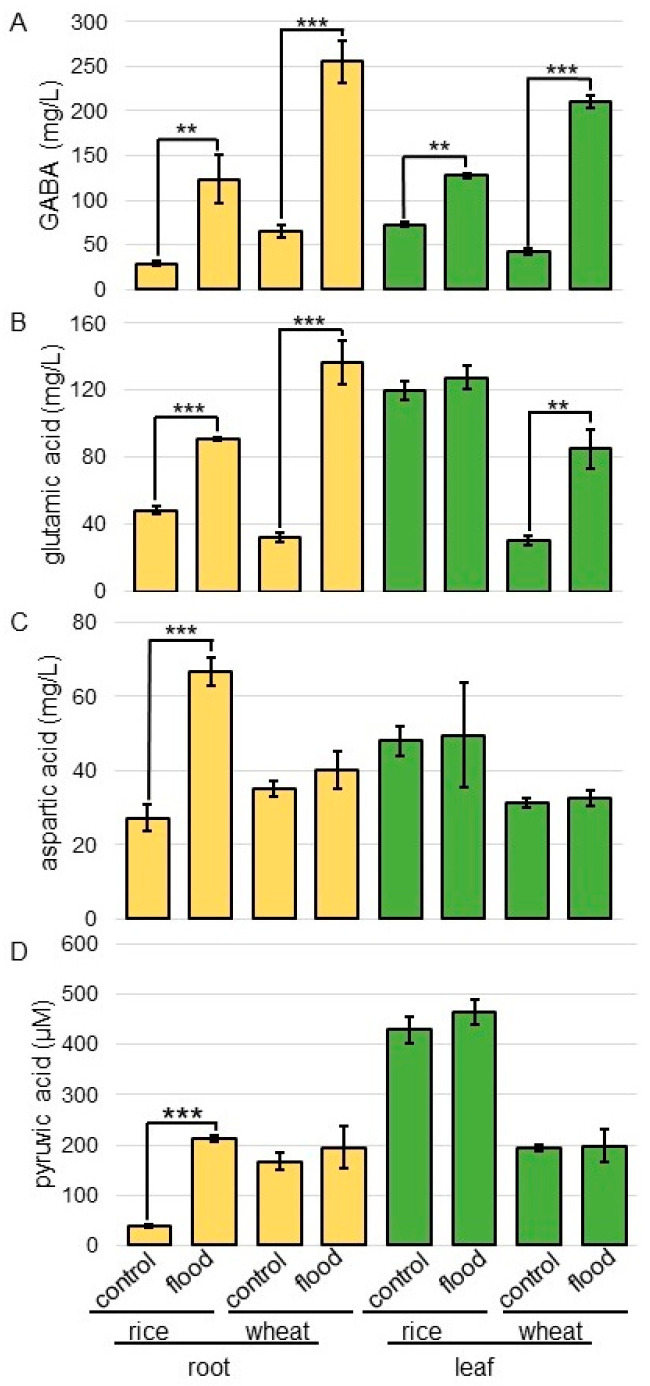
The contents of GABA, Glu, pyruvic acid, and Asp in rice and wheat under flooding stress. Three-day-old rice and wheat were flooded with or without water for 2 days. Root and leaf were homogenized with phosphate-buffered saline. After removal of proteins, the contents of GABA (**A**), Glu (**B**), pyruvic acid (**C**), and Asp (**D**) were analyzed. The data are presented as mean ± SD from three independent biological replicates. Asterisks indicate significant changes in the relative intensity of signal band in the flooding compared to control according to Student’s *t*-test (***, *p* < 0.001; **, *p* < 0.01).

**Figure 8 ijms-25-05229-f008:**
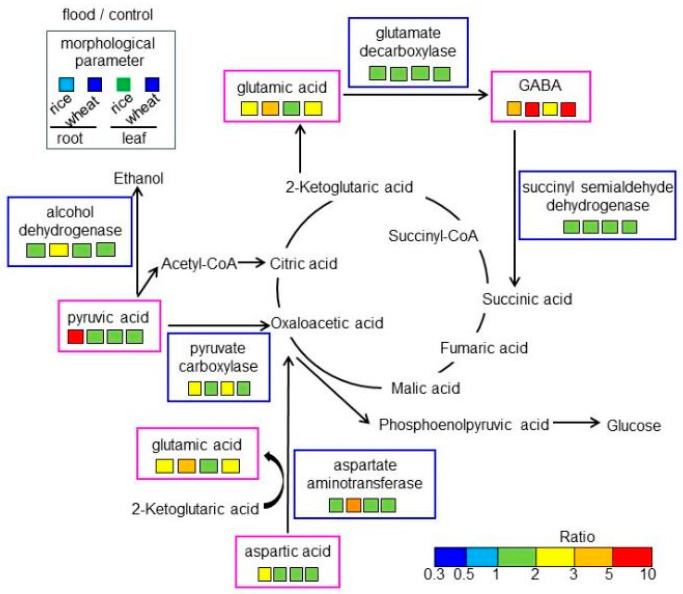
A mapping of altered amino acids and proteins to amino-acid metabolism in rice and wheat under flooding stress. Identified amino acids and proteins were mapped onto a pathway according to the KEGG database. Differences in color indicate the different ratio ranges of the quantities of metabolites, which were calculated by dividing the content in flooded rice or wheat by the content in corresponding non-flooded plants. Additionally, the ratio of morphological parameters is shown on the top left. Each set of four boxes shows “flood/control in rice root”, “flood/control in wheat root”, “flood/control in rice leaf”, and “flood/control in wheat leaf”.

## Data Availability

Data is contained within the article and Appendix A.

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
