# Peer review of "The Changes of Amino-Acid Metabolism between Wheat and Rice during Early Growth under Flooding Stress"

_ijms, 2024, doi:10.3390/ijms25105229_

Round 1

Reviewer 1 Report

Comments and Suggestions for Authors

The manuscript by Komatsu et al. considers the influence of flooding on morphology and amino-acid and protein metabolism in wheat and rice. The investigation seems to be interesting. However, I have some questions and remarks.

(1)    Were the plants completely submerged in water during the experiment? Could the plants (especially, wheat) be died in the water?

(2)    Does Figure 3 show data for a shoot or a root?

(3)    What are the perspectives of the investigation?

(4)    Please, improve the colorbars in Figures 4 and 8. In Figure 4 only 8 colors are used for designation of 9 variants of ratio. In Figure 8 only 4 colors used for designation of 5 variants of ratio. These points confuse interpretation of result of investigation.

(5)    Please, add schema of influence of amino-acid and protein metabolism on morphologic parameters in wheat and rice.

Author Response

Reviewer 1

The manuscript by Komatsu et al. considers the influence of flooding on morphology and amino-acid and protein metabolism in wheat and rice. The investigation seems to be interesting. However, I have some questions and remarks.

(1) Were the plants completely submerged in water during the experiment? Could the plants (especially, wheat) be died in the water?

Answer: Thank you very much for your question. In this study, rice and wheat were completely submerged (flooded) for two days.  In the case of wheat, there was no reference about early-stage flooding, although there was a report about the flooding stress of 3-week-old plant. The sentence commented by reviewer has been corrected based on the original reference in red. The corrected sentence is as follow: “Rice grows well in standing water, but most varieties will die if completely submerged for more than 3 days [7]”. Reference 7 is “Emerick, K.; Ronald, P.C. Rice: engineering rice for climate change. Cold Spring Harb. Perspect. Biol. 2019, 11, a034637”.

(2) Does Figure 3 show data for a shoot or a root?

Answer: Thank you very much for the comment. Figure 3 shows data for a root. The material used in Figure 3 and Figure 4 has been clarified in the legend of Figure 3 and the result section with red colour.

(3) What are the perspectives of the investigation?

Answer: Based on the morphological results, the roots were more affected compared to leaves by flooding stress.  The roots of rice and wheat were used for amino-acid analysis. The reason why amino-acid analysis was adopted in this study is that changes in the number of amino acids in plants under flooding stress are related to various metabolisms. For example, amino acids related to glycolysis, including the phosphoglycerate family (Ser and Gly), shikimate family (Phe, Tyr and Trp), and pyruvate family (Ala, Leu and Val), are greatly elevated. Members of the Asp family (Asn, Lys, Met, Thr and Ile) and the Glu family (Glu, Pro, Arg and GABA) were accumulate as well [19]. This explanation has been added in the text in red. Reference 19 is “Yemelyanov, V.V.; Puzanskiy, R.K.; Shishova, M.F. Plant Life with and without Oxygen: A Metabolomics Approach. Int. J. Mol. Sci. 202324, 16222”.

(4) Please, improve the colorbars in Figures 4 and 8. In Figure 4 only 8 colors are used for designation of 9 variants of ratio. In Figure 8 only 4 colors used for designation of 5 variants of ratio. These points confuse interpretation of result of investigation.

Answer: As suggested, colorbars have been improved. Differences in color indicate the different ratio ranges of the quantities of metabolites, which was calculated by dividing the content in flooded rice or wheat by the content in corresponding non-flooded plants. This explanation has been specified in the legends of Figures 4 and 8. For example, red color in Figure 4 indicates a change of 20 times or more but less than 30 times.

(5) Please, add schema of influence of amino-acid and protein metabolism on morphologic parameters in wheat and rice.

Answer: Thank you very much for your suggestion. In Figure 8, the morphological parameters have been added and the legend of this figure has been modified.

Reviewer 2 Report

Comments and Suggestions for Authors

The present manuscript describe effect of flooding stress on zoung seedlings of rice and wheat. The topic is very actual and interesting and some part of results are signigicant. However,  there are some clarity requiry regarding design of research and data interpretation.

Title do not reflect contents of the paper.   Authors studied only GABA and some aminoacid metabolism. This must be reflected in title, not general biochemical analysis.

Abstract.

Authors come to conclusion thet flooding stress inhibit wjeat growth through GABA metabolism, but did niot present evidences. GABA meatbolism accompanied this effect, but may not be the reason of such inhibition.

Line 18: Asparatic and pyrivic acid can not increased itself, contents or metabolism can increase.

 Introduction:

Line 28: Flooding did not directly reduce soil quality.

Lines 37 – 38: not clear what do authors mean.

Line 45: why soaybean?

Line 69 – 70: the most sensitive to fllooding stress is hormone metabolism. While pyruvate and other aminoacid also may affected. Authors need to discuss hormone metabolism as well.

Line 79: morphology can not be measure, moprhological parametrs can be.

Results.

Authors need to “clean” rice seeds before soaking to accelerate water uptake. It will meake design more clear.

Fig 1 and lines 272- 273 are contradictory. 2 days old seedlings and 3 days stress or 3 days old seedlings and 2 days stress??

Moreover, you menation 5 cm water above sand surface. What was coleptile length at this time point? Did seedlings formed true leaf before flloding? Is it the same stage for rice and wheat?

Figure 2: no scale bar. It will be great to label a panle (A, B, C etc).  Kinetics must be shown as well: seedlings before flood and after flood  (with scale bar).

Did authors study root in details. Shorter/smaller root mean either short proliferation zone with less cells, or absence/reduce cell elongation with more similar cell number. This is a key question here.

Line 106: why did you choose aminoacid metabolism in root? Root is well-know results of polar auxin transport and canalisation. This is primary mecahnsim of roort growth. How aminoacid metabolism linked with auxin canalization/transport?

Line 119: do you mean contents in the root? Aminoacid itself can not increase itself! Moreover, based on previous question: what do you mean nmol/ml? How this related with root fresh weight? And did AA contents increase per cell or because of flood prevent cell expansion and similar FW have more cell??

Figure 4: which type oft he cell have such metabolism? Root have a mature cell, meristemic cell, root hairs tec. Each cell type (and position) have own metabolism. Did aminoacid you measure derived from root or transported from the shoots?

Fig 7 vs fig 3: what do you mean as mg/L in fig 7? Why nmol/ml as in figure 3?

Discussion: flooding led to epigenetic changes, BUT this is not reflected in the text.

Moreover, one of the effect of flloding is photosynthesis. Inhibition of photosynthesis led to carbon startvation and immediately stop root grotwh. To prove this author can check exogenous sucrose application under flloding stress.

Some language corrections are required.

Comments on the Quality of English Language

Some polishing are required

Author Response

Reviewer 2

The present manuscript describe effect of flooding stress on zoung seedlings of rice and wheat. The topic is very actual and interesting and some part of results are signigicant. However,  there are some clarity requiry regarding design of research and data interpretation.

Title do not reflect contents of the paper.   Authors studied only GABA and some aminoacid metabolism. This must be reflected in title, not general biochemical analysis.

Answer: We appreciate your point. Title has been corrected as follows: “The Changes of Amino-Acid Metabolism between Wheat and Rice during Early Growth under Flooding Stress”

Abstract.

Authors come to conclusion thet flooding stress inhibit wjeat growth through GABA metabolism, but did niot present evidences. GABA meatbolism accompanied this effect, but may not be the reason of such inhibition.

Answer: We are sorry for this mistake. The conclusion parts of the abstract section have been corrected as follows: “These results suggest that flooding stress significantly inhibits wheat growth through upregulating amino-acid metabolism and increasing alcohol-fermentation system compared to rice. When the aerobic-metabolic system is activated by the inhibition of plant growth, GABA content increases.”

Line 18: Asparatic and pyrivic acid can not increased itself, contents or metabolism can increase.

 Introduction:

Answer: We are sorry for these mistakes. Based on your kind point, this mistake has been adjusted in all parts of this manuscript.

Line 28: Flooding did not directly reduce soil quality.

Answer: This sentence has been corrected based on the original reference. The corrected sentence is as follows: “Climate change, which is one of the challenges facing modern agriculture, is exacerbated by global population growth and soil-quality deterioration [1]”. Reference 1 is “Kopeć, P. Climate change-The rise of climate-resilient crops. Plants 202413, 490”.

Lines 37 – 38: not clear what do authors mean.

Answer: This sentence has been corrected as follows: “Because flooding stress is based on initial fluctuations and also induces secondary changes, its mechanism is not completely understood.”

Line 45: why soaybean?

Answer: Thank you very much for your comment. This sentence has been rewritten using the rice reference as follows.: SUB1A, which is an ethylene-response factor (ERF), confers submergence tolerance to rice by restricting shoot elongation during the inundation period. it is proposed to limit shoot growth by regulating gibberellic acid signaling [9]. Reference 1 is “Schmitz, A.J.; Folsom, J.J.; Jikamaru, Y.; Ronald, P.; Walia, H. SUB1A-mediated submergence tolerance response in rice involves differential regulation of the brassinosteroid pathway. New Phytol. 2013, 198, 1060-1070.”

Line 69 – 70: the most sensitive to fllooding stress is hormone metabolism. While pyruvate and other aminoacid also may affected. Authors need to discuss hormone metabolism as well.

Answer: As suggeted, the explanation of hormonal metabolism has been added in the section of introduction in red as follows.: “Plants can cope with flooding conditions by embracing an orchestrated set of morphological adaptations and physiological adjustments, which are regulated by an elaborate hormonal signaling network (Wang and Komatsu, 2022).  In wheat seedlings, the upregulation of TDCYUC1, and PIN9 involved in auxin biosynthesis and transport contributed to high levels of auxin, which was required for nodal root induction during hypoxia (Nguyen et al., 2018). Regarding salicylic acid enhancement of wheat tolerance to waterlogging, it was revealed that salicylic acid promoted formation of axile roots independent from ethylene, but its effect on adventitious rooting was dependent on ethylene (Koramutla et al., 2022). The hormonal signaling networks are also important in wheat under flooding stress, but they are less clearly understood than in rice.“

Line 79: morphology can not be measure, moprhological parametrs can be.

Answer: We are sorry for this mistake. The word “morphology” has been corrected as “morphological parameters”.

Results.

Authors need to “clean” rice seeds before soaking to accelerate water uptake. It will meake design more clear.

Answer: As suggested, Figure 1 has been revised with detailed experimental information and figure legend has been improved.

Fig 1 and lines 272- 273 are contradictory. 2 days old seedlings and 3 days stress or 3 days old seedlings and 2 days stress??

Answer: We are sorry for this mistake. This sentence has been corrected as follows.: “To induce flooding stress, water was added 5 cm above the sand surface, and 3-day-old plants were soaked for 2 days.”

Moreover, you menation 5 cm water above sand surface. What was coleptile length at this time point? Did seedlings formed true leaf before flloding? Is it the same stage for rice and wheat?

Answer: We are sorry that coleoptile length was not measured on 3-day-old seedling. The first leaf of rice and wheat was already formed and their seedling sizes were about the same before the flood. However, the rice was slightly smaller, although rice seeds were allowed to absorb water in a Petri dish for 7 days before sowing. These sentences have been added in the section of results in red.

Figure 2: no scale bar. It will be great to label a panle (A, B, C etc).  Kinetics must be shown as well: seedlings before flood and after flood  (with scale bar).

Did authors study root in details. Shorter/smaller root mean either short proliferation zone with less cells, or absence/reduce cell elongation with more similar cell number. This is a key question here.

Answer: As suggested, the photographs of seedlings before and after flood were added with scale bar with additional experiments. Furthermore, the label of panel has been added with A, B, C, D, and E. Based on this correction, figure legends and the section of result has been corrected in red. We are sorry that we could not measure short proliferation zone with less cells or absence/reduce cell elongation with more similar cell number, because the size of plant was very small.

Line 106: why did you choose aminoacid metabolism in root? Root is well-know results of polar auxin transport and canalisation. This is primary mecahnsim of roort growth. How aminoacid metabolism linked with auxin canalization/transport?

Answer: Thank you very much for your question and comments. Based on the morphological results, the roots of rice and wheat were used for amino-acid analysis. The relationship between amino-acid metabolism and auxin has been added in the discussion section with references in red as follows: Amino acids and auxin need to be transported across cell membranes to exert their functions in various organs, which relies on specific carrier proteins on the all membrane, mainly amino-acid transporter proteins (Williams and Miller, 2001). Two families of amino-acid transporter proteins were identified, which are amino acid/auxin permease and amino-acid polyamine choline gene family (Saier et al., 2016). Amino acid/auxin permease protein is an enzyme, which mediates the movement of a variety of amino acids and auxin into and out of cells (Saier et al., 2009), and participates in regulating the transmembrane structure of amino acids and long-distance transport of amino acids in the body, as well as other life processes (Tegeder, 2012). Under saline-alkali stress, amino acid/auxin permease were differentially expressed between salt-alkali tolerant millet variety and salt-alkali sensitive millet variety (Wang et al., 2024). 

References:

Williams, L.E.; Miller, A.J. Transporters responsible for the uptake and partitioning of nitrogenous solutes. Annu. Rev. Plant Physiol. Plant Mol. Biol. 2001, 52, 659–688.

Saier, M.H. Jr.; Reddy, V.S.; Tsu, B.V.; Ahmed, M.S.; Li, C.; Moreno-Hagelsieb G.  The transporter classification database (TCDB): Recent advances. Nucleic Acids Res. 2016, 44, D372–D379.

Saier, M.H. Jr.; Yen, M.R.; Noto, K.; Tamang, D.G.; Elkan, C. The transporter classification database: Recent advances. Nucleic Acids Res. 2009, 37, 274–278.

Tegeder M. Transporters for amino acids in plant cells: Some functions and many unknowns. Curr. Opin. Plant Biol. 2012, 15, 315–321.

Wang, H.; Li Y.; Guo, Z.; Zhou, X.; Zhao, Y.; Han, Y.; Lin, X. Genome-wide identification of AAAP gene family and expression analysis in response to saline-alkali stress in foxtail millet (Setaria italica L.). Sci Rep. 2024, 14, 3106. 

Line 119: do you mean contents in the root? Aminoacid itself can not increase itself!

Answer: We are sorry for this mistake. Based on your kind point, this mistake has been corrected in all parts of this manuscript.

Moreover, based on previous question: what do you mean nmol/ml? How this related with root fresh weight? And did AA contents increase per cell or because of flood prevent cell expansion and similar FW have more cell??

Answer: We are sorry that we could not explain it clearly. In this study, root with 500 mg of fresh weight was homogenized in 500 μL. “nmol” shows amino-acid concentration per mg of root-fresh weight. As suggested, this explanation has been added in the section of materials and methods in red.

Figure 4: which type oft he cell have such metabolism? Root have a mature cell, meristemic cell, root hairs tec. Each cell type (and position) have own metabolism. Did aminoacid you measure derived from root or transported from the shoots?

Answer: We are sorry that we cannot answer for this comment. In this study, we did not take into account the movement of amino acids from above-ground to below-ground parts, so we analyzed changes in amino acids in the underground roots.  In the next stage of this research, we would like to conduct research with a view to conducting experiments that can answer the requests reviewer has proposed.

Fig 7 vs fig 3: what do you mean as mg/L in fig 7? Why nmol/ml as in figure 3?

Answer: We are sorry that we could not explain it clearly, again. In this study for Fig. 7, root with 250 mg of fresh weight was homogenized in 250 μL. “mg/L” shows metabolites weight (mg) per homogenates (L) from root-fresh weight or leaf-fresh weight (250 mg). This explanation has been added in the section of materials and methods in red.

Discussion: flooding led to epigenetic changes, BUT this is not reflected in the text.

Answer: As suggested, the explanation of epigenetic changes under flooding stress has been added in the section of introduction in red as follows: “Epigenetic changes are associated with abiotic stress in crop. Heat stress increased DNA methylation levels in heat-sensitive rice but decreased it in heat-tolerant rice (Li et al., 2022). The total DNA methylation in sesame increased under drought stress but decreased under flooding stress (Komivi et al., 2018). In wheat, DNA demethylation significantly increased waterlogging-related gene expression in tolerant genotypes under hypoxic stress (Pan et al., 2020). Both waterlogging and half submergence increased the total DNA methylation level, but this was decreased under full submergence in wheat (Li et al., 2023). However, the potential mechanisms of epigenetic regulation under flooding stress remain largely unknown in rice and wheat.”

References:

Li, B.; Cai, H.Y.; Liu, K.; An, B.Z.; Wang, R.; Yang, F.; Zeng, C.L.; Jiao, C.H.; Xu, Y.H. DNA methylation alterations and their association with high temperature tolerance in rice anthesis. J. Plant Growth Regul. 202242, 780–794.

Komivi, D.; Marie, A.M.; Zhou, R.; Zhou, Q.; Yang, M.; Ndiaga, C.; Diaga, D.; Wang, L.H.; Zhang, X.R. The contrasting response to drought and waterlogging is underpinned by divergent DNA methylation programs associated with transcript accumulation in sesame. Plant Sci. 2018277, 207–217. 

Pan, R.; Xu, Y.H.; Xu, L.; Zhou, M.X.; Jiang, W.; Wang, Q.; Zhang, W.Y. Methylation changes in response to hypoxic stress in wheat regulated by methyltransferases. Russ. J. Plant Physiol. 202067, 323–333.

Li, B.; Hua, W.; Zhang, S.; Xu, L.; Yang, C.; Zhu, Z.; Guo, Y.; Zhou, M.; Jiao, C.; Xu, Y. Physiological, epigenetic, and transcriptome analyses provide insights into the responses of wheat seedling leaves to different water depths under flooding conditions. Int. J. Mol. Sci. 202324, 16785.

Moreover, one of the effect of flloding is photosynthesis. Inhibition of photosynthesis led to carbon startvation and immediately stop root grotwh. To prove this author can check exogenous sucrose application under flloding stress.

Answer: Based on the comments from reviewer, the experiment has been performed. However, morphological parameters did not change by additional sucrose. For this experiment, the concentration of sucrose may be important. Although we could not find suitable concentration of sucrose now, we will try to this experiment in future.

Some language corrections are required.

Answer: Based on comments from reviewer, this article has been corrected by native English speaker.

Round 2

Reviewer 1 Report

Comments and Suggestions for Authors

The authors fully responded to my comments

Author Response

Reviewer 1

The authors fully responded to my comments

Answer: Thank you very much for your improvement for this article.

Reviewer 2 Report

Comments and Suggestions for Authors

The text is much better now. Please, make small further corrections.

You performed 7 days incubation of rice seeds. why not to scarify seeds??Please, look:  Awan TH, Chauhan BS, Cruz PCSta (2014) Influence of Environmental Factors on the Germination of Urena lobata L. and Its Response to Herbicides. PLoS ONE 9(3): e90305. doi:10.1371/journal.pone.0090305

Lines 333-335 and 367 - 369: the classical samples:buffer ratio is 1:10, but you can use 1:1. This may give not correct results.

Comments on the Quality of English Language

minor polishing (space between words etc).

Author Response

Reviewer 2

The text is much better now. Please, make small further corrections. You performed 7 days incubation of rice seeds. why not to scarify seeds??Please, look:  Awan TH, Chauhan BS, Cruz PCSta (2014) Influence of Environmental Factors on the Germination of Urena lobata L. and Its Response to Herbicides. PLoS ONE 9(3): e90305. doi:10.1371/journal.pone.0090305

Answer: Thank you very much for your question. In the case of article by Awan et al. (2014), seeds were scarified with 98% sulfuric acid for 30 min.  Seeds were washed with running tap water for 5 min after treating with sulfuric acid and before being subjected to various treatments.  In our case, wheat seeds are disinfected with sodium hypochlorite and washed with water. In this study, the same method was used for rice as for wheat. When using this method, the seeds used in this study fully germinated and were in uniform condition by the second day. After 7 days of absorbing water, when rice was sown, its growth condition was the same as that of wheat (Figure 2A). Of course, a great deal of preliminary experimentation was required to align the growth periods, but we ultimately decided to use these conditions. As suggested, the legend of Figure 2 has been added the sentence about the preparation of rice seeds in blue.

Lines 333-335 and 367 - 369: the classical samples:buffer ratio is 1:10, but you can use 1:1. This may give not correct results.

Answer: We apologize for not providing sufficient explanation. We have re-calculated the data are correct. 

Comments on the Quality of English Language

minor polishing (space between words etc).

Answer: We are sorry this problem. This article has been carefully revised now I blue.